# Identifying and prioritising unanswered research questions for people with hyperacusis: James Lind Alliance Hyperacusis Priority Setting Partnership

Kathryn Fackrell [1,2] Linda Stratmann,[1] Veronica Kennedy,[3] Carol MacDonald,[4] Hilary Hodgson,[1] Nic Wray,[1] Carolyn Farrell,[1] Mike Meadows,[1] Jacqueline Sheldrake,[5] Peter Byrom,[6] David M Baguley [1,2,7] Rosie Kentish,[1] Sarah Chapman,[8] Josephine Marriage,[9] John Phillips,[10] Tracey Pollard,[11] Helen Henshaw [1,2] Toto A Gronlund,[12] Derek J Hoare [1,2]

For numbered affiliations see end of article.

**Correspondence to**
Dr Kathryn Fackrell;
kathryn.fackrell@nottingham.ac.uk

## ABSTRACT

**Objective**  To determine research priorities in hyperacusis that key stakeholders agree are the most important.

**Design/setting**  A priority setting partnership using two international surveys, and a UK prioritisation workshop, adhering to the six-staged methodology outlined by the James Lind Alliance.

**Participants**  People with lived experience of hyperacusis, parents/carers, family and friends, educational professionals and healthcare professionals who support and/or treat adults and children who experience hyperacusis, including but not limited to surgeons, audiologists, psychologists and hearing therapists.

**Methods**  The priority setting partnership was conducted from August 2017 to July 2018. An international identification survey asked respondents to submit any questions/uncertainties about hyperacusis. Uncertainties were categorised, refined and rephrased into representative indicative questions using thematic analysis techniques. These questions were verified as 'unanswered' through searches of current evidence. A second international survey asked respondents to vote for their top 10 priority questions. A shortlist of questions that represented votes from all stakeholder groups was prioritised into a top 10 at the final prioritisation workshop (UK).

**Results**  In the identification survey, 312 respondents submitted 2730 uncertainties. Of those uncertainties, 593 were removed as out of scope, and the remaining were refined into 85 indicative questions. None of the indicative questions had already been answered in research. The second survey collected votes from 327 respondents, which resulted in a shortlist of 28 representative questions for the final workshop. Consensus was reached on the top 10 priorities for future research, including identifying causes and underlying mechanisms, effective management and training for healthcare professionals.

**Conclusions**  These priorities were identified and shaped by people with lived experience, parents/carers and healthcare professionals, and as such are an essential resource for directing future research in hyperacusis. Researchers and funders should focus on addressing these priorities.

### Strengths and limitations of this study

► This is the first time patients and healthcare professionals in hyperacusis have worked together to identify priorities for research.
► Used established James Lind Alliance methodology to systematically and transparently identify top research priorities.
► Survey responses from key stakeholders all over the world, which led to representative priorities for the UK, but with applicability internationally.
► Some final indicative questions are broad and may be open to interpretation.

## INTRODUCTION

Hyperacusis is a hearing disorder involving an increased sensitivity or decreased tolerance to sound at levels that would not trouble most individuals. It differs from recruitment (a narrowing of the auditory dynamic range due to hearing loss), misophonia (an acquired aversive reaction to specific human generated sounds) and phonophobia (persistent, abnormal and unwarranted fear of sound).[1–3] For the person experiencing hyperacusis, everyday sounds of various loudness, such as the sound of the domestic appliances or electric hand dryers, can be uncomfortable or painful, and are perceived much louder and more intense than they are. This can be frightening and overwhelming and can cause anxiety and distress. Hyperacusis occurs as a primary complaint but is also commonly associated with other conditions including tinnitus, head trauma, depression, dementia, post-traumatic stress syndrome and fibromyalgia.[4] Current estimates suggest that hyperacusis is experienced by 3.7% of children,[5] and up to 9.2% of adults in a general population,[6]

with only 2% formally diagnosed by a physician.[7] Prevalence is particularly high in certain populations, for example, 65% of children and adults with Asperger's syndrome or autism spectrum disorder,[8 9] and 95% of children and young adults with Williams syndrome experience hyperacusis.[10 11]

Often chronic and disabling, hyperacusis can have a significant effect on daily life. For example, it can interfere with education, the ability to work and/or participation in social and family life. In a recent cohort study involving 357 adult patients, 25 distinct domains of hyperacusis-associated problems were identified, including fear, avoidance behaviours, pain, activity limitations and impairments to quality of life such as reduced ability to work and socialise.[12] For children with hyperacusis, school environments (including lunch rooms and classrooms) can be challenging places, and strategies need to be in place to ensure that their well-being and educational needs are met.[13] People experiencing hyperacusis often become isolated and lose independence, avoiding situations where noise levels are outside their control. This affects the individual and their family or carers. Many thousands of people experiencing hyperacusis or their family members turn to support and information provided online through Facebook, forums and other social media.[14]

Hyperacusis remains medically unexplained with no definitive diagnosis, aetiology or cure.[4] For clinicians, the focus is on helping people manage their symptoms, not 'cure' their hyperacusis, although no formal clinical practice guidance currently exists. The structure and provision of services for hyperacusis vary across the UK, with a variety of different medical specialties (including ear, nose and throat, audiology, audiovestibular medicine, clinical psychology) involved in the management of people experiencing hyperacusis. Clinical management in general includes non-standardised assessment using sound-based tolerance tests, and questionnaires to determine and measure the specific effects hyperacusis has on an individual's life.[15] Approaches to treatment include sound therapy, counselling and cognitive–behavioural therapy[16] but there is little or no evidence of their effectiveness. Indeed, despite its prevalence and clinical implications, hyperacusis research is in its infancy and there are many controversies in the field.[17–19] For example, only two randomised controlled trials specifically testing hyperacusis treatments have been conducted,[16] and the only systematic review on hyperacusis examines prevalence in children and adolescents.[20] With such an open field, it is essential to identify and address research priorities that are immediately relevant and important to those affected by hyperacusis, and those who provide care for them. A number of different approaches exist, with some such as Child Health Nutrition Research Initiative method, James Lind Alliance (JLA) Method and Combined Approach Matrix, having well-defined structure.[21] In the UK, the JLA, a non-profit initiative hosted by the National Institute of Health Research (NIHR), offers one of most

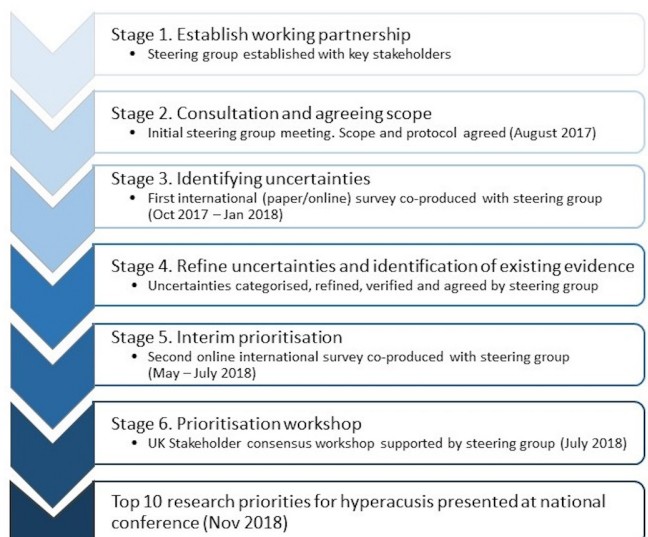

**Figure 1** The six-stage methodological process for the hyperacusis priority setting partnership.

established and pragmatic processes for prioritising health research questions. The JLA guiding principles are to bring together patients, carers and healthcare professionals in priority setting partnerships (PSP) to identify and agree priorities for research. The JLA provides established step-by-step methodology to manage and complete a PSP, and ensures that all processes are accountable and transparent.

The aim of this study was (1) to identify the questions about hyperacusis that are important to people with lived experience, parents/carers, family, educational professionals and healthcare practitioners involved in the care of people who have hyperacusis and (2) to prioritise these questions according to the relevance and importance to these groups to direct future research and funding in hyperacusis.

## METHODS

The hyperacusis PSP took place over an 11-month period between August 2017 and July 2018. The six-stage methodology adhered to the PSP process prescribed by the JLA (figure 1). Documentation produced for the PSP, including the protocol, posters and initial survey, is published on the JLA website (http://www.jla.nihr.ac.uk/priority-setting-partnerships/hyperacusis/).

### Stage 1: establishing a working partnership
The research leads (KF and DJH) from NIHR Nottingham Biomedical Research Centre (BRC), with support from our user organisation representative (LS), initiated the PSP and appointed a steering group to oversee and contribute to the PSP process. The steering group included people with lived experience of hyperacusis, a parent of a child living with hyperacusis, members of organisations who support those with hyperacusis and healthcare professionals (audiologists, clinical psychologists, ear, noise and

throat and audiovestibular medicine representatives) with experience of working with and supporting those (adult and children) with hyperacusis (online supplemental appendix 1). The researchers (KF, DH, HHe) were allowed to attend the steering group meetings but did not participate in the prioritisation process. In order to reach a wide range of stakeholders, a small number of relevant partner organisations were identified through steering group members with established links (online supplemental appendix 2). A JLA adviser (TAG) acted as neutral facilitator for the PSP, providing support and guidance throughout the whole PSP process, ensuring that there were equal contributions from all key stakeholders and that at all times the PSP was conducted in a fair and transparent way. The PSP coordinator and information specialist (KF) managed the surveys and data, communication and performed the analysis and literature searches. The steering group directed and advised on all stages of the PSP, including the scope and relevant target stakeholders (figure 1).

### Patient and public involvement

Patient and public involvement is a key component of the JLA process. People with lived experience and parents/carers of those with hyperacusis were included in the steering group for the PSP. As steering group members, they were actively involved in all aspects of the project, including coproducing all materials, undertaking thematic analysis on subsets of submitted uncertainties and reviewing all documentation. The dissemination strategy, developed with the steering group, included engagement with members of the public through plain English summaries, charity magazine articles and an Evidently Cochrane blog coproduced with our patient representatives (https://www.evidentlycochrane.net/hyperacusis-what-need-to-know/).

### Stage 2: consultation and agreeing scope

The steering group raised concerns over the numerous definitions associated with hyperacusis. Definitions in the literature include but are not limited to '*decreased sound tolerance and difficulty adjusting to sudden shifts in volume*',[22] '*a heightened awareness of sound*'[23] or '*hypersensitivity to sound*',[24] '*intolerance to everyday sounds that causes significant distress and impairment in social, occupational, recreational and other day-to-day activities*'[25] and '*negative reactions following exposure to sound that would not evoke the same response in an average listener*'.[26] Patient representatives passionately felt that the definition needed to state that it was a hearing condition, and that a description of the experience had to be provided. Therefore, using evidence from a recent scoping review[16] and the experience of steering group members, we defined hyperacusis as '*a hearing disorder involving an increased sensitivity or decreased tolerance to sound at levels that would not trouble most individuals. For the person experiencing hyperacusis, everyday sounds can be unpleasant, intense, frightening, painful and overwhelming and can cause anxiety and distress*'.

All uncertainties related to hyperacusis for both adults and children were considered within scope. For example, the scope included uncertainties about the (1) causes, (2) assessment, (3) management (treatment, rehabilitation, self-management, interventions) of hyperacusis; (4) prevention and education; (5) care and support and (6) healthcare services for hyperacusis. To be inclusive and ensure a representative range of hyperacusis uncertainties were identified, the steering group made the decision that the initial survey and interim prioritisation would be available internationally. However, due to funding restrictions, it was agreed that the final prioritisation workshop would involve UK-based representatives only.

### Stage 3: identification of potential research questions

An online and paper version of the survey were coproduced with the steering group. The survey consisted of an explanation of the study and what was required, six open questions relating to each element of the scope (eg, What question(s) about the *how hyperacusis is assessed and/or diagnosed* would you like to see answered by research?) with example questions (eg, How do clinicians check for (xxx) in children?) and an open general question (Do you have any other questions that you feel are important for researchers to answer but do not fall in the areas above?). Participants were instructed that each question was optional and that they could write as much or as little as they liked for each. Participants were also asked to complete an optional demographic information section (stakeholder group, age, gender, country of residence, ethnic background, healthcare professionals' main profession and any other hearing conditions). Participants answered the survey anonymously and were instructed to contact the study coordinator (KF) via a study-specific email address if they wanted to be involved in the next stages of the study.

A pilot phase was used with the steering group and researchers at NIHR Nottingham BRC (including a public and patient involvement manager) to check that the survey was written clearly, understandable and easy to complete using the online survey software (www.surveymonkey.com). The text in the survey was refined to provide clarity on the number of questions that participants had to complete (ie, each question was optional). The survey was circulated via steering group members (including to the hyperacusisresearch.net international network based in the USA), partner organisation contact lists, promotion in newsletters and national conferences, printed posters in clinics and through dedicated social media channels (including to the international membership of the Hyperacusis Support and Research Facebook group which is led by the UK and US administrators). In addition to submitted uncertainties, we also searched for any research uncertainties highlighted in reviews of hyperacusis.

## Stage 4: refining research questions and identification of existing literature

The aim of this stage was to categorise and refine uncertainties, verify them as uncertainties and create a list of indicative questions for the next stage. A qualitative thematic analysis approach was taken to categorise and refine uncertainties, and generate a smaller number of representative indicative questions. This involved (1) data familiarisation (reading and rereading submissions), (2) developing codes for uncertainties, (3) identifying potential categories and subcategories, and finally (4) reviewing and refining final categories and subcategories. Based on the categorisation, the uncertainties were rephrased into representative indicative research questions. The analysis was undertaken by the information specialist with input from steering group members throughout. The decision about whether a submission was out of scope was initially decided by the information specialist and subsequently reviewed by the steering group. Submissions were excluded based on criteria set by the steering group before the analysis was conducted. The criteria was any uncertainties (1) outside of scope (eg, uncertainties about misophonia or recruitment alone), (2) advice or information requests and (3) broad or ambiguous submissions. Because a previous PSP on tinnitus had ranked a research priority on the relationship between tinnitus and hyperacusis in their top 10,[27] the steering group decided that any submissions related to tinnitus would also be excluded. Where more than one uncertainty was identified in the same submission, the response was duplicated and assigned to the appropriate category. The final categories and refined questions with the original submitted uncertainties were reviewed by all steering group members to ensure that the final indicative questions were understandable and true representations of the original uncertainties. The steering group agreed the final list of indicative questions.

The verification process involved a review of current evidence for each indicative question to determine whether the questions were 'true uncertainties' or had already been answered by research. Given the broad scope, a comprehensive search approach was taken to ensure that all available evidence for indicative questions was identified. The Cochrane Ear, Nose and Throat Disorders Group Trials Register; the Cochrane Central Register of Controlled Trials; PubMed; Embase; International Standard Randomised Controlled Trial Number registry; ClinicalTrials.gov; Google Scholar and Google were searched from inception of each question to May 2018 for systematic reviews, past and current trials, scoping reviews and clinical guidelines (including guidelines produced for hearing and tinnitus to check whether there was any evidence presented for hyperacusis) relating to the indicative questions. All evidence was recorded for each indicative question with links to journal papers. Indicative questions were only considered 'true uncertainties' where there was no evidence available, or was of low quality (ie, likely further evidence would change the estimate of the effect), or where the available systematic reviews needed updating or indicated continuing uncertainties. The steering group reviewed all evidence and agreed a final list of questions which were included in the interim prioritisation survey.

## Stage 5: interim prioritisation

The purpose of this stage was to reduce the long list of questions to a shorter manageable by asking a wide range of people with lived experience of hyperacusis, and professionals to prioritise the indicative questions they considered most important. An online survey was created for this purpose using an interactive card-sorting software (www.optimalworkshop.com). The refined list of indicative questions was presented in a randomised sequence of cards for each participant on the screen. Participants were asked to read all the questions and select (drag and drop) the 10 questions that they thought were most important for research, in any order they wished (they were not asked to rank the questions). To help with the process of identifying their top 10 questions, an intermediate 'questions of interest' box was included as a holding pen for questions they wished to consider. Participants were informed that they could swap and change questions from the boxes as often as they liked. Before taking part in the card sorting, participants were required to specify the stakeholder group(s) that applied to them. Once again, all survey responses were anonymous. Optional demographic information was collected (age, gender, healthcare professionals' main profession) and participants were invited to register their interest to attend the prioritisation workshop via email (online supplemental appendix 3). The survey was piloted with the steering group and researchers from NIHR Nottingham BRC to check that all instructions were clear and understandable, and to confirm the usability of the software. Instructions were refined to ensure clarity on the minimum number of questions to include in the 'top 10' box. The survey was circulated as an open invitation that was not restricted to those who had completed the first survey. The invitation was also sent to all people who had registered an interest in participating in the next stage following the initial survey, and circulated via steering group members, partner organisation contact lists, promotions in newsletters and at tinnitus support groups, printed posters in clinics and through dedicated social media channels.

All indicative questions that were moved to the top 10 box were ranked based on the number of summed responses for each stakeholder group. For the analysis, weighted quantitative methods were used, in which the summed responses from professionals were weighted based on the number of responses from people with lived experience. This method was used, and agreed by the steering group, to ensure that the final indicative questions for the workshop were representative of all groups (ie, of equal significance to votes from across all stakeholder groups). The results of the analysis were reviewed. The steering group also decided that if questions that

captured under-represented groups (mental health, autism, dementia, learning difficulties) were not represented in the top set, then these questions would be taken forward to the final workshop and the decision was made to take a top set of approximately 25–30 indicative questions to the workshop.

## Stage 6: final prioritisation workshop

A 1-day workshop with representatives from each stakeholder group was held at the University of Nottingham in July 2018. The purpose of the workshop was to establish consensus on the top 10 research priorities for hyperacusis through a blend of whole group and subgroups discussions facilitated by independent JLA advisers with no experience of hyperacusis. The workshop was chaired by the JLA adviser for this PSP (TAG). Remote participation to the final workshop was considered by the steering group but determined not feasible due to volume-control issues for representatives living with hyperacusis and small/large group discussion format. Therefore, only UK-based representatives were invited to participate in the final workshop. Healthcare and educational professionals, people with lived experience of hyperacusis and parents of children living with hyperacusis were invited to participate via contact details submitted following the two surveys, organisation contact lists and social media. Participants were selected with the aim of 50:50 representation for 'lived experience' (including adults, children and parents) and professionals. A small number of steering group members were encouraged to participate in the final workshop. Attending steering group members ensured all participants were supported with information on the PSP processes and any other support as required.

Using a modified nominal group process, participants were divided into three groups with mixed representatives from people with lived experience and professionals. Each group worked together initially discussing their own personal priorities. In the second round, cards with the indicative questions printed on them were laid out for the group to view, loosely ordered to reflect the discussion in the first round. The group were tasked to work together to rank the questions from most important to least. On the back of the cards were the results from the interim prioritisation survey to assist with decision-making, if needed. The rankings from each group were combined, and the membership of the groups was changed so that each group now had someone from one of the other two groups (again with mixed representatives in each). These new small groups were all presented the combined rankings from the previous round, and tasked to review and adjust the ranking to reflect the consensus of the group. The rankings from each group were again combined and finally, the whole group came together to discuss and adjust this combined ranking, until whole group consensus was reached on the top 10 priorities for hyperacusis.

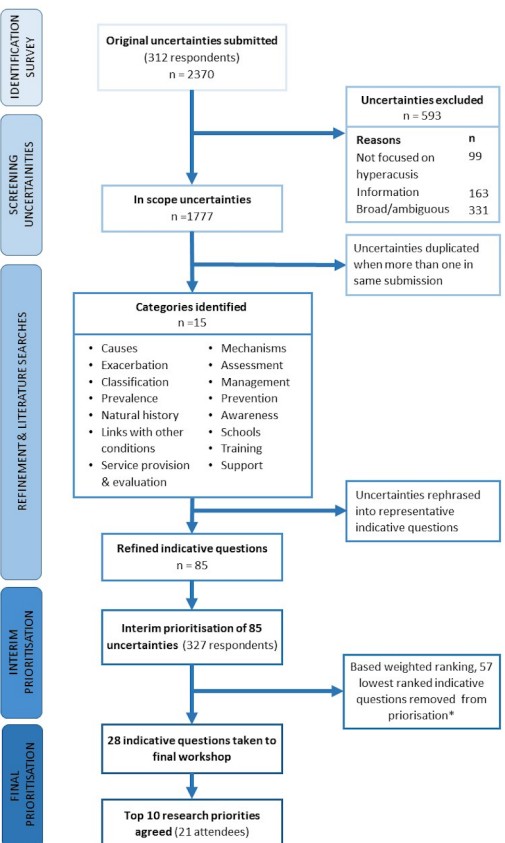

**Figure 2** Flow diagram of prioritisation process from submitted research questions to final prioritisation. *One additional question on dementia with lower ranking was included at final workshop as per *a priori* criteria set by steering group.

## RESULTS

Figure 2 shows a flow diagram of the study.

### Demographics of initial survey respondents

Two-thousand, three-hundred and seventy uncertainties about hyperacusis were submitted, online and on paper, by 312 respondents, from October 2017 to January 2018. The number of uncertainties submitted per respondent ranged from 1 to 37, with the average number submitted per respondent being 10. Respondents included people with lived experience (58%), healthcare professionals from a range of disciplines (28%), parents/carers (7%) and family and friends (1%) of those with hyperacusis. A small number of educational professionals (1%) also submitted uncertainties on the school environment and children living with hyperacusis (table 1). Several respondents also identified themselves as having an additional role, including respondents with lived experience of hyperacusis who also identified themselves as a parent (n=1), healthcare professional (n=1) or educational professional (n=1) (table 1). Respondents were mainly residents in the UK (65%), America (13%) and Ireland (4%), with remaining residing in various countries in Europe, Asia and Africa. Demographics of the survey respondents are presented in table 1. No additional

**Table 1** Demographics of respondents to the identification survey and interim prioritisation survey

| Characteristics | Identification n | Interim prioritisation n |
|---|---|---|
| **Role** | | |
| Person with hyperacusis | 181 | 218 |
| *Tinnitus* | 74 | — |
| *Hearing loss* | 36 | — |
| *Tinnitus and hearing loss* | 66 | — |
| Parent of a child with hyperacusis | 19 | 14 |
| Carer of someone with hyperacusis | 2 | 4 |
| Friend or family member of someone with hyperacusis | 3 | 4 |
| Education professionals (eg, school teacher, lecturer) | 3 | 2 |
| Healthcare professional | 87 | 82 |
| *Audiologist* | 43 | 38 |
| *Paediatric audiologist* | 7 | 2 |
| *ENT* | 2 | 6 |
| *Audiovestibular physician* | 11 | 7 |
| *Clinical scientist* | 3 | 5 |
| *Hearing therapist* | 9 | 3 |
| *Speech-language pathologist* | — | 2 |
| *Speech and language therapist* | — | 1 |
| *Clinical psychologist* | 4 | |
| *Psychiatrist* | 1 | 1 |
| *Clinical hypnotherapist* | 1 | — |
| *Intensive care physician* | — | 1 |
| *GP* | — | 1 |
| *Paediatrician* | 1 | 2 |
| *Nurse* | — | 3 |
| *Dentist* | — | 1 |
| *Chiropractor and nutritionist* | 1 | — |
| More than one role identified | 7 | 34 |
| Other | 8 | 3 |
| Not specified | 9 | — |
| **Age** | | |
| Under 10 | 0 | 4 |
| 10–20 | 7 | 11 |
| 21–30 | 19 | 30 |
| 31–40 | 40 | 49 |
| 41–50 | 67 | 37 |
| 51–60 | 85 | 59 |
| 61–70 | 57 | 42 |
| 71–80 | 16 | 19 |
| 80+ | 1 | 2 |
| Not specified | 20 | 74 |
| **Sex** | | |
| Female | 194 | 95 |

Continued

**Table 1** Continued

| Characteristics | Identification n | Interim prioritisation n |
|---|---|---|
| Male | 103 | 152 |
| I would describe it in another way | — | 2 |
| Not specified | 5 | 78 |
| **Ethnicity** | | |
| White | 276 | — |
| Asian | 8 | — |
| Mixed/multiple ethnic group | 4 | — |
| Arab | 1 | — |
| Black | 1 | — |
| Prefer not to say | 20 | — |
| Other (not specified) | 5 | — |
| **Country of residence** | | |
| UK | 203 | — |
| USA | 42 | — |
| Ireland | 11 | — |
| Australia | 6 | — |
| Canada | 6 | — |
| New Zealand | 4 | — |
| Sweden | 4 | — |
| Netherlands | 3 | — |
| Germany | 2 | — |
| South Africa | 2 | — |
| Austria | 1 | — |
| Belgium | 1 | — |
| Brazil | 1 | — |
| Denmark | 1 | — |
| Finland | 1 | — |
| France | 1 | — |
| Greece | 1 | — |
| Portugal | 1 | — |
| Philippines | 1 | — |
| Saudi Arabia | 1 | — |
| Switzerland | 1 | — |

ENT, ear, nose, and throat; GP, general practitioner.

questions were identified in the research literature that were not already covered by the uncertainties submitted.

### Refinement of research questions and identification of existing literature

Of the 2370 submitted uncertainties, 593 were removed as out of scope. The remaining 1777 uncertainties were subjected to thematic analysis and generated 15 categories (figure 2). Each category grouped at least two uncertainties submitted by more than one respondent from the stakeholder groups. Subcategories were generated for the larger categories (causes, management and links with

other conditions) to facilitate the process of rephrasing uncertainties into representative indicative questions (online supplemental appendix 4). The grouped uncertainties were refined, and 85 indicative questions were formulated as representative of the original uncertainties. As a result of discussions, the steering group decided to leave some indicative questions as generic in terms of age (ie, specifying children or adults). This would allow for researchers to address these from both angles and therefore increasing the number of possible priorities addressed. The existing literature did not provide any high-quality evidence that addressed any of the indicative questions; therefore, all 85 indicative questions were included in the interim prioritisation.

### Interim priorities determined from second survey

The interim prioritisation survey opened in May 2018 and closed in July 2018. Three-hundred and twenty-seven respondents voted for their top 10 indicative research questions. Respondents were people with lived experience (67%), healthcare professionals (25%) and then a small number of parents/carers (6%), family and friends (1%), and educational professionals (1%). Demographics of the survey respondents are presented in table 1. As expected, there was some imbalance in the priorities of different stakeholder groups. However, the weighted ranked scores for all 85 questions were reviewed by the steering group and the top 27 indicative questions were identified as representative of the votes received by all stakeholder groups. For example, 18 of the top 25 questions voted for by people with lived experience were included in the top 27. It was agreed, as per the *a priori* criteria set by the steering group, that the top 27 indicative questions, and an additional indicative question on dementia, were all priorities (table 2).

### Top 10 priorities

The 1-day workshop held in July 2018 included 21 participants, representing people with lived experience of hyperacusis (n=6) (including two teachers), parents of children living with hyperacusis (n=5) and healthcare professionals (n=10) from a range of disciplines that care for and support those with hyperacusis (including audiologists, a paediatric clinical psychologist, clinical psychologists and an audiovestibular physician). Following this workshop, consensus was reached on a top 10 research priorities for hyperacusis (table 3). These questions cover a broad range of topics in hyperacusis including causes, mechanisms, prevalence, management and treatment, and healthcare provider knowledge and training.

### Publicity

The top 10 priorities were disseminated to all participants involved in the process via a contact email address or social media and shared by steering group members and partner organisations through their contact lists. The top 10 priorities were unveiled at the British Academy of Audiology national conference in November 2018,[28]

published in audiology and charity magazine articles[29 30] and in a published letter in a scientific journal.[31] The list of original submitted uncertainties, the 85 indicative questions and the final list of priorities taken to the workshop have been made available on and promoted through the JLA website (http://www.jla.nihr.ac.uk/priority-setting-partnerships/hyperacusis/).

## DISCUSSION

The hyperacusis PSP has successfully brought together people with lived experience of hyperacusis and healthcare professionals in hyperacusis to identify and agree priorities for research. These priorities were therefore identified and shaped by people with lived experience, parents/carers and healthcare professionals. Although this process did not identify radically different questions, it did highlight the sheer number of unanswered questions about hyperacusis. The top 10 priorities reflect the lack of evidence and guidance on the assessment and management of hyperacusis, including the need for training in hyperacusis. These priorities highlight the need for research to focus on the underlying mechanisms and types of hyperacusis, and prevalence of hyperacusis in specific populations. The list of priorities identified here will therefore be of interest to researchers and funders of hearing/health research, and will hopefully direct future research and funding calls.

There are many notable strengths to this study. First, people with lived experience and parents/carers were involved throughout the process, from the initial planning for the JLA with our user organisation representative, planning and overseeing the project as members of the steering group, and participation in both surveys worldwide, and in the final workshop. All of this enabled people with lived experience and parents/carers to have the opportunity for a voice and to influence the future priorities for research on hyperacusis. Second, this PSP also took an inclusive approach to gathering the uncertainties and interim voting by having two international online surveys, and although the final consensus on the priorities was reached with UK representatives only, the top 10 priorities, and the longer list of priorities, have international relevance as they represent unanswered questions submitted by people with lived experience, parents/carers and healthcare professionals from around the world. However, finer grain analysis of the initial submissions by country was not meaningful because there were only a small number of submissions from most countries. Predominantly, responses were received from participants in the UK and USA, which is partly a product of the promotional routes taken in the online surveys. However, one notable difference was observed, that the initial submissions by US participants frequently asked about reimbursement of healthcare costs, whereas for the UK this is not an issue. Therefore, while it is possible that some health service provision and evaluation questions are less relevant in some countries, the remaining indicative

**Table 2**  The top 28 ranked questions from interim prioritisation survey

| Rank | Research question | Summed responses for each stakeholder group | | |
| --- | --- | --- | --- | --- |
| | | Person with hyperacusis | Parents/ carers/family | Prof~ |
| 1 | What is the most effective treatment approach for hyperacusis in adults? | 60 | 6 | 80 |
| 2 | Which treatment approaches are most effective for different types or severities of hyperacusis? | 51 | 6 | 77 |
| 3 | Does avoidance of sound improve hyperacusis or make it worse? | 60 | 8 | 51 |
| 4 | Is hyperacusis related to physical changes in the ear or brain? | 69 | 3 | 46 |
| 5 | Is hyperacusis due to physical or psychological issues or is it a combination of both? | 41 | 7 | 65 |
| 6 | Which psychological therapy (eg, counselling, cognitive–behavioural therapy, mindfulness) is most effective for hyperacusis? | 18 | 7 | 85 |
| 7 | What are the 'red flags' for serious underlying conditions in hyperacusis? | 28 | 7 | 68 |
| 8 | Which criteria should be met to diagnose hyperacusis in adults/children? | 25 | 5 | 71 |
| 9 | What area(s) of the brain and patterns of activity is/are associated with hyperacusis? | 51 | 4 | 46 |
| 10 | What is the most effective treatment approach for hyperacusis in children? | 10 | 7 | 83 |
| 11 | What are the risk factors for developing hyperacusis or making it worse? | 37 | 3 | 54 |
| 12 | What is the best way of using sound in therapy for hyperacusis? | 20 | 2 | 71 |
| 13 | What is the essential knowledge/training required for health professionals to appropriately refer or effectively manage hyperacusis? | 27 | 5 | 57 |
| 14 | What management approach for hyperacusis is most effective for adults/ children with autism? | 8 | 5 | 74 |
| 15 | Is there an association between hyperacusis and other ear-related conditions (eg, superior canal dehiscence syndrome, Meniere's, Waardenburg syndrome, vertigo, vestibular migraines)? | 36 | 3 | 43 |
| 16 | What is the best way to differentiate hyperacusis from other hearing conditions (eg, recruitment, misophonia, Meniere's, tinnitus)? | 30 | 3 | 48 |
| 17 | Which self-help interventions are effective for hyperacusis? | 40 | 7 | 34 |
| 18 | Is hyperacusis linked to other sensitivities/conditions? | 34 | 11 | 34 |
| 19 | Does nerve damage cause the pain associated with hyperacusis? | 60 | 5 | 11 |
| 20 | What is the prevalence of hyperacusis in a general population and other specific populations (eg, people with autism, mental health issues, learning disabilities, hearing loss)? | 17 | 3 | 54 |
| 21 | Which drugs are effective for hyperacusis? | 45 | 3 | 26 |
| 22 | Can noise exposure cause hyperacusis (or make it worse)? | 59 | 6 | 9 |
| 23 | Which interventions in a school setting are useful for children with hyperacusis? | 11 | 7 | 54 |
| 24 | Would restoring hearing (eg, regenerating nerve cells) improve hyperacusis? | 56 | 5 | 6 |
| 25 | Are there different meaningful types of hyperacusis? | 23 | 3 | 40 |
| 26 | What is the relationship between mental health and hyperacusis? | 20 | 6 | 40 |
| 27 | What care is most effective for recent onset/acute hyperacusis? | 30 | 1 | 34 |
| 28 | What is the association between hyperacusis and dementia?* | 16 | 0 | 9 |

*Additional question on dementia with lower ranking included at final workshop as per *a priori* criteria set by steering group.
~, weighted professionals score; Prof, healthcare and educational professionals.

questions collected in this PSP should be considered unanswered priorities for all. Third, there were notable differences in the interim prioritisation between people with lived experience and healthcare professionals. For example, healthcare professionals were more likely to prioritise questions relating to effective treatments for both adults and children and differentiating hyperacusis, whereas people with lived experience were more likely to

**Table 3** Hyperacusis priority setting partnership top ten priorities for future research

| Priority order | Research question |
|---|---|
| 1 | What is the most effective treatment approach for hyperacusis in children? |
| 2 | What is the prevalence of hyperacusis in a general population and other specific populations (eg, people with autism, mental health issues, learning disabilities, hearing loss)? |
| 3 | Are there different meaningful types of hyperacusis? |
| 4 | What is the essential knowledge/training required for health professionals to appropriately refer or effectively manage hyperacusis? |
| 5 | Which treatment approaches are most effective for different types or severities of hyperacusis? |
| 6 | Is hyperacusis due to physical or psychological issues or is it a combination of both? |
| 7 | Which psychological therapy (eg, counselling, cognitive–behavioural therapy, mindfulness) is most effective for hyperacusis? |
| 8 | What management approach for hyperacusis is most effective for adults/children with autism? |
| 9 | What is the best way of using sound in therapy for hyperacusis? |
| 10 | Which self-help interventions are effective for hyperacusis? |

prioritise causes before effective treatments and differentiating hyperacusis (table 2). However, by using weighted ranking, the top 10 reflected the mixed priorities from all stakeholder groups, of which five were in the top 10 for people with lived experience and parents/carers during the interim prioritisation and six were in the top 10 for professionals during the interim prioritisation. Finally, by using the JLA process for prioritisation, independent JLA advisors supported and guided the whole PSP process, including managing any potential conflicts arising from differences in perspectives and power relations between healthcare professionals and people with lived experience of hyperacusis and ensuring that all stakeholders had an equal voice.

There were some challenges with the PSP process. One challenge was interpreting the submissions that were more narrative stories than specific research questions. Due to the nature of the PSP, many submissions were asking more than one question within a narrative story, especially submissions from people experiencing hyperacusis who until this point had not been asked their point of view. To ensure that all vital information and possible questions from the narratives were identified and recorded, the information specialist worked closely with steering group members to interpret the narratives. Another challenge arose in planning the JLA final face-to-face workshop; people who experience hyperacusis find that travelling and participating in large and small group discussions can lead to pain, discomfort or distress. In order to overcome any foreseeable problems with recruiting people with lived experience, different options were discussed by the steering group. Remote participation was suggested as a potential avenue for people to take part in the workshop. However, it was decided from experience of remote steering group meetings, and advice sought from the JLA, that online discussion facilities such as Skype or teleconferences would not overcome the problems with discomfort, would be physically and mentally taxing to participants, and could impede the discussion process.

The venue for the workshop was discussed. It was decided that the workshop would take place at the University of Nottingham during the summer period when there are no students on campus as this would reduce the number of potential problems with loud noises. The facilities used for the workshop were quiet rooms with limited external sounds. Participants were given the option to travel the day before the workshop to allow time to recover from travelling. All facilitators and participants were aware of potential problems that could occur and a dedicated quiet room was available for anyone to go to for a break. The lead researchers were on hand to offer any additional support to participants. Therefore, although a relatively small number of people with lived experience participated in the final workshop (six people with lived experience and five parents of children with hyperacusis), it did not impede the discussions or consensus as there was balanced representation from each stakeholder group. Also, the results from interim prioritisation, where a large number people experiencing hyperacusis voted, informed the discussions.

Many of the priorities highlighted by this PSP clearly mark areas that are less well researched in hyperacusis. For example, in 2014, a two-part review of hyperacusis research[17 18] highlighted key areas in hyperacusis that were seen as important to progress, including but not limited to, properly defining and subtyping hyperacusis, identifying and understanding mechanisms and the role of the brain, and evaluating current treatments (particularly sound therapy and drugs). These topics also appeared in reviews published in 2017[16] and 2018.[19] These more recent reviews also reported a need for up-to-date prevalence figures and valid measurement tools. All these topics reflect the 28 priority questions identified in our interim prioritisation survey. This would suggest that the majority of recently published research in hyperacusis has not necessarily been addressing the areas of concern. A search of Google Scholar revealed that the most recent publications for hyperacusis have focused on basic

research questions around causes of hyperacusis,[32–34] or on research investigating hyperacusis and the workplace.[35] The only relevant ongoing trial is investigating 'Exposure Therapy for Auditory Sensitivity in Autism' ( ClinicalTrials.gov Identifier: NCT03206996). Although this research is important, it is somewhat disheartening to observe that the majority of the areas highlighted by a review in 2014 still remain an issue now. This prioritisation exercise highlighted the need for research to have a much greater focus on clinical needs (management approaches) and effective treatments, as well as aspects of underlying mechanisms and subtyping. With these questions now in place, it is a responsibility to people with lived experience and healthcare professionals that researchers in the field pursue these questions as their priorities, and thereby minimise research waste in this field.[36] All sources of data related to these questions, whether through clinical audit, retrospective analyses, undergraduate or postgraduate projects or bespoke grant applications, will usefully start to address these questions and build capacity in this neglected field. It will be important that researchers engage with funders to formulate feasible projects they are likely to endorse.

This PSP highlighted a potential challenge within current healthcare for hyperacusis. A larger number of patients engaged in each of the surveys than healthcare professionals. This could be a reflection of our current healthcare system, in which only a small number of healthcare professionals have experience in such a specialised subject as hyperacusis. Moreover, it was apparent from the list of priorities that identifying the essential training needs of healthcare professionals for each clinical specialty involved with hyperacusis is a priority. Furthermore, the number of out-of-scope uncertainties asking for information on hyperacusis and care and support further highlighted this problem and a clear need for greater awareness of hyperacusis. Current knowledge and awareness of hyperacusis in the UK are sadly lacking. Therefore, informational resources are needed to raise awareness of hyperacusis and provide guidance on seeking care and support. However, in order to provide this information, first service evaluations need to be conducted to provide a clear picture of the current position of hyperacusis care and identify the areas of need.

## CONCLUSION

Hyperacusis is a topic of growing interest. Here, we have identified a set of priority questions that reflect what people with lived experience, parents/carers and healthcare professionals want answered by research. As such, researchers and funders should focus on addressing these priorities.

**Author affiliations**

[1]National Institute of Health Research Nottingham Biomedical Research Centre, University of Nottingham, Nottingham, UK

[2]Hearing Sciences, Division of Clinical Neuroscience, School of Medicine, University of Nottingham, Nottingham, UK

[3]Department of Paediatric Audiology, Bolton NHS Foundation Trust, Bolton, UK

[4]Department of Psychology, University of Stirling, Stirling, UK

[5]The Tinnitus and Hyperacusis Centre, London, UK

[6]Peter Byrom Audiology, Sheffield, UK

[7]Nottingham Audiology Services, Nottingham University Hospitals NHS Trust, Nottingham, UK

[8]Cochrane UK, Oxford, UK

[9]Children's Hearing Evaluation & Amplification Resource, London, UK

[10]Department of Otolaryngology, Norfolk and Norwich University Hospital NHS Trust, Norwich, UK

[11]Action on Hearing Loss, London, UK

[12]The James Lind Alliance, National Institute of Health Research Evaluation, Trials and Studies Coordinating Centre, Southampton, UK

**Acknowledgements** The authors wish to acknowledge the people with lived experience of hyperacusis, parents of children living with hyperacusis, families and friends, carers and educational and healthcare professionals who submitted responses to the international surveys. In addition, they would like to thank the partner organisations (Supplemental appendix 2) who supported and promoted this work, Sandra Smith for checking the literature searches and providing support at the final prioritisation workshop, the JLA for support and guidance throughout the process and all attendees and JLA facilitators (Katherine Cowan, Sheela Upadhyaya, and Toto Anne Gronlund) at the final workshop who worked tirelessly to achieve a consensus on the top 10 research questions.

**Contributors** KF and DJH made the initial application to the JLA for a hyperacusis PSP. KF organised steering group meetings and teleconferences, contacted and enrolled all partner organisations, designed the surveys, liaised with organisations throughout the PSP, promoted the PSP through a dedicated Twitter account, conducted thematic analysis and literature reviews, organised and enrolled participants for the final workshop, and drafted the final manuscript. DJH organised venues and facilities for meetings and final workshop, drafted and circulated minutes of every meeting and teleconference, conducted initial thematic analysis and analysis on the interim prioritisation survey data. TAG was the JLA independent facilitator for the PSP, chaired all steering group meetings and the final workshop, provided support and guidance throughout the whole PSP process to members of the steering group and the research team including reviewing documentation. LS, VK, CM, HHo, NW, CF, MM, JS, PB, DMB, RK, SC, JM, JP, TP and HHe attended steering group meetings, made decisions on the scope, documentation and analysis to identify indicative questions for survey and final workshop; they codeveloped the information presented in the surveys and promotions, commented and rephrased the 85 indicative questions and analysed and identified the 28 indicative questions for the final workshop. LS, VK, CM, CF, HHo, JM, DMB, RK, and JS conducted thematic analysis on a subset of submitted uncertainties. SC provided advice on promotional routes and promoted the PSP through Evidently Cochrane blog. All authors read and approved the final manuscript for submission.

**Funding** This work was supported through grants from the British Society of Audiology and Action on Hearing Loss. The Partnership was supported by the National Institute for Health Research Nottingham Biomedical Research Centre.

**Disclaimer** KF is funded by the National Institute for Health Research (NIHR Post-Doctoral Fellowship, Dr Kathryn Fackrell, PDF-2018–11-ST2-003). HHe is funded by the National Institute for Health Research (NIHR Career Development Fellowship, Dr Helen Henshaw, CDF-2018–11-st2-016). DJH is funded by the National Institute for Health Research (NIHR) Biomedical Research Centre programme. However the views expressed in this publication are those of the author(s) and not necessarily those of the NHS, the National Institute for Health Research or the Department of Health and Social Care.

**Competing interests** DJH is vice-chair of the British Society of Audiology. KF is a member of the British Society of Audiology Tinnitus and Hyperacusis special interest group. DMB is president of the British Tinnitus Association. NW is an employee of the British Tinnitus Association.

**Patient consent for publication** Not required.

**Ethics approval** This work did not require ethics approval as per the JLA guidance and guidance published by the NHS National Patient Safety Agency National Research Ethics Service.

**Provenance and peer review** Not commissioned; externally peer reviewed.

**Data availability statement** Data are available in a public, open access repository.

**ORCID iDs**
Kathryn Fackrell http://orcid.org/0000-0001-6529-8643
David M Baguley http://orcid.org/0000-0002-0767-0723
Helen Henshaw http://orcid.org/0000-0002-0547-4403
Derek J Hoare http://orcid.org/0000-0002-8768-1392

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
