## [Reviewer comments · BMJ Open]

ARTICLE DETAILS

TITLE (PROVISIONAL)	Identifying and prioritising unanswered research questions for people with hyperacusis: the James Lind Alliance Hyperacusis Priority Setting Partnership
AUTHORS	Fackrell, Kathryn; Stratmann, Linda; Kennedy, Veronica; MacDonald, Carol; Hodgson, Hilary; Wray, Nic; Farrell, Carolyn; Meadows, Mike; Sheldrake, Jacqueline; Byrom, Peter; Baguley, David; Kentish, Rosie; Chapman, Sarah; Marriage, Josephine; Phillips, JS; Pollard, Tracey; Henshaw, Helen; Gronlund, Toto; Hoare, Derek

VERSION 1 – REVIEW

REVIEWER	Richard Morley Cochrane United Kingdom I was a James Lind Alliance Adviser from April 2014 to March 2017
REVIEW RETURNED	25-Jun-2019

GENERAL COMMENTS	I have reviewed the manuscript and regard this as a very thorough description of the prioritisation process undertaken by this JLA Priority Setting Partnership. It thoroughly and clearly sets out the methods that were followed in undertaking the PSP and a critical exploration of the strengths and weakness of the process. I have few comments to make as the paper efficiently addresses the description of the methods. I would have been interested to hear some mention of the following: (i) Whether the partnership considered other priority setting methodologies that exist, and strengths and weaknesses of different processes. For example why not a patient only priority setting process? Why this model as opposed to, say, Dialogue Model etc., However I do recognise the high profile and pragmatic nature of the JLA process. (ii) The international ambitions of the PSP are interesting and some more description and critical analysis of this would have been interesting to me (how did the PSP reach out to a wider geography, what were the challenges, why did those countries respond, what were the similarities/differences between submitted uncertainties). (iii) Greater analysis of the top 10 and top 28 questions would have been interesting (and ultimately why the PSP undertook its work though I do accept that there is some consideration given to this). For example, what themes emerged, what similarities and differences were there between health care professional priorities, and people with lived experience/carers? (iv) a key feature of the JLA PSP is bringing together HCPs and people with lived experience. How were the differences in
--

	perspectives and power relations managed? The answers to (i) and (iv) above might strongly relate (for example, language, resources, support for patient/carer participants, the way the meetings were managed). There is a developing body of critical literature in this area. Having said all of the above, I think this is a thorough and well expressed description of the process, important for promoting further research in the condition, and I would be delighted to see it published.
--	---

REVIEWER	Dan Hasson Karolinska Institutet, Sweden
REVIEW RETURNED	18-Jul-2019

GENERAL COMMENTS	This study is impressively well-written with a profound design as a foundation. The aim was to determine research priorities in hyperacusis that key stakeholders agree are the most important. The results indicate that research that is conducted about hyperacusis is not really in line with the needs and expectations of different stakeholders. Therefore, this study is important. I only have a couple of minor suggestions for improvements:  1. The tables are difficult to read and need more supporting texts to make them more pedagogic. It is not easy to understand what the figures in the tables refer to and the headings of the columns are a bit ambiguous. 2. The discussion would clearly benefit from a section about practical implications of the findings. How should these results be used and by whom? What are the next steps in order to ensure research funding for covering the identified priorities. What are your clear messages to scientists, clinicians and patients? This study identifies the gaps and should, to my view, also describe ideas about how to close them. Otherwise this knowledge is of limited importance.
---

VERSION 1 – AUTHOR RESPONSE

Reviewer 1.

Comment 1: I would have been interested to hear some mention of the following:

(i) Whether the partnership considered other priority setting methodologies that exist, and strengths and weaknesses of different processes. For example why not a patient only priority setting process? Why this model as opposed to, say, Dialogue Model etc. However, I do recognise the high profile and pragmatic nature of the JLA process.

We have revised the manuscript to include examples of other prioritisation processes that were considered and further justification for the selecting the JLA process.

“With such an open field, it is essential to identify and address research priorities that are immediately relevant and important to those affected by hyperacusis, and those who provide care for them. A number of different approaches exist, with some such as Child Health Nutrition Research Initiative (CHNRI) method, James Lind Alliance Method and Combined Approach Matrix, having well-defined structure [21]. In the UK, The James Lind Alliance (JLA) a non-profit initiative hosted by the National

Institute of Health Research (NIHR), offers one of most established and pragmatic processes for prioritising health research questions.” [pg.5]

(ii) The international ambitions of the PSP are interesting and some more description and critical analysis of this would have been interesting to me (how did the PSP reach out to a wider geography, what were the challenges, why did those countries respond, what were the similarities/differences between submitted uncertainties).

We have updated the methods to include examples of the methods used to engage with people with lived experience of hyperacusis internationally and updated the discussion section to include information on the challenges with the international submissions. The partner organisation which include international membership are listed in the supplementary information.

“The survey was circulated via steering group members (including to the hyperacusisresearch.net international network based in the US), partner organisation contact lists, promotion in newsletters and national conferences, printed posters in clinics, and through dedicated social media channels (including to the international membership of the Hyperacusis Support & Research Facebook group which is led by UK and US administrators).” [pg. 7-8]

“However, finer grain analysis of the initial submissions by country was not meaningful because there were only a small number of submissions from most countries. Predominantly responses were received from participants in the UK and US, which is partly a product of the promotional routes taken in the online surveys. However, one notable difference was observed, that the initial submissions by US participants frequently asked about reimbursement of healthcare costs, whereas for the UK this is not an issue. Therefore, whilst it is possible that some health service provision and evaluation questions are less relevant in some countries, the remaining indicative questions collected in this PSP should be considered unanswered priorities for all.” [pg. 13-14]

(iii) Greater analysis of the top 10 and top 28 questions would have been interesting (and ultimately why the PSP undertook its work though I do accept that there is some consideration given to this). For example, what themes emerged, what similarities and differences were there between health care professional priorities, and people with lived experience/carers?

We have revised the discussion section considering the similarities and differences between healthcare professionals and people with lived experiences interim priorities.

“Thirdly, there were notable differences in the interim prioritisation between people with lived experience and healthcare professionals. For example, healthcare professionals were more likely to prioritise questions relating to effective treatments for both adults and children and differentiating hyperacusis, whereas people with lived experience were more likely to prioritise causes then effective treatments and differentiating hyperacusis (Table 2). However, by using weighted ranking, the top 10 reflected the mixed priorities from all stakeholder groups, of which five were in the top 10 for people with lived experience and parents/carers during the interim prioritisation and six were in the top 10 for professionals during the interim prioritisation.” [pg. 14]

(iv) a key feature of the JLA PSP is bringing together HCPs and people with lived experience. How were the differences in perspectives and power relations managed? The answers to (i) and (iv) above might strongly relate (for example, language, resources, support for patient/carer participants, the way the meetings were managed). There is a developing body of critical literature in this area. Having said all of the above, I think this is a thorough and well expressed description of the process, important for promoting further research in the condition, and I would be delighted to see it published.

Very many thanks for your positive endorsement of this work. We have updated the manuscript to include a statement on the JLA advisor's role in managing potential conflicts.

"Finally, by using the JLA process for prioritisation, independent JLA advisors supported and guided the whole PSP process, including managing any potential conflicts arising from differences in perspectives and power relations between healthcare professionals and people with lived experience of hyperacusis and ensuring that all stakeholders had an equal voice." [pg 14]

Reviewer 2:

Comment 1. The tables are difficult to read and need more supporting texts to make them more pedagogic. It is not easy to understand what the figures in the tables refer to and the headings of the columns are a bit ambiguous.

We have simplified all the tables and clarified headings.

Comment 2. The discussion would clearly benefit from a section about practical implications of the findings. How should these results be used and by whom? What are the next steps in order to ensure research funding for covering the identified priorities. What are your clear messages to scientists, clinicians and patients? This study identifies the gaps and should, to my view, also describe ideas about how to close them. Otherwise this knowledge is of limited importance.

"With these questions now in place it is a responsibility to people with lived experience and healthcare professionals that researchers in the field pursue these questions as their priorities, and thereby minimise research waste in this field [36]. All sources of data related to these questions, whether through clinical audit, retrospective analyses, undergraduate or postgraduate projects, or bespoke grant applications, will usefully start to address these questions and build capacity in this neglected field. It will be important that researchers engage with funders to formulate feasible projects they are likely to endorse." [pg 15]

Very many thanks for your positive endorsement of this work.